

# Genome-wide identification of the *PEBP* genes in pears and the putative role of *PbFT* in flower bud differentiation

Shuliang Zhao, Yarui Wei, Hongguang Pang, Jianfeng Xu, Yingli Li, Haixia Zhang, Jianguang Zhang and Yuxing Zhang

College of Horticulture, Hebei Agricultural University, Baoding, Hebei, China

## ABSTRACT

Although *Phosphatidylethanolamine-binding protein* (*PEBP*) genes have been identified in several plants, little is known about *PEBP* genes in pears. In this study, a total of 24 *PEBP* genes were identified, in which 10, 5 and 9 were from *Pyrus bretschneideri* genome, *Pyrus communis* genome and *Pyrus betuleafolia* genome, respectively. Subsequently, gene structure, phylogenetic relationship, chromosomal localization, promoter regions, collinearity and expression were determined with these *PEBP* genes. It was found that only *PbFT* from *PEBP* genes of *P. bretschneideri* was relatively highly expressed in leaves during flower bud differentiation. Whereas, expression patterns of *TFL1* homologues, *gene23124* and *gene16540*, were different from *PbFT* in buds. The expression pattern and the treatment of reduction day-length indicated that the expression of *PbFT* in leaves were regulated by day-length and circadian clock. Additionally, the phenotype of transgenic *Arabidopsis* suggested that *PbFT* played a role in not only promoting flower bud differentiation, but also regulating the balance between vegetative and reproductive growth. These results may provide important information for further understanding of the evolution and function of *PEBP* genes in pears.

# INTRODUCTION

Phosphatidylethanolamine-binding proteins (PEBPs) are named for their evolutionarily conserved phosphatidylethanolamine-binding domains and widely found in plants, animals and yeast (*Chautard et al., 2004*; *Leeggangers et al., 2018*; *Sun et al., 2018*). In plants, the *PEBP* genes family is mainly divided into three subfamilies, *FLOWERING LOCUS T* (*FT*)-like, *TERMINAL FLOWER 1* (*TFL1*)-like and *MOTHER OF FT AND TFL1* (*MFT*)-like (*Li et al., 2015*). *MFT*-like is the ancestor of the *FT*-like and *TFL1*-like subfamilies and is not found in moss and lycopodium. *FT/TFL1* homologous genes appeared along with the evolution of seed plants. *FT/TFL1*-like gene plays an important role in the transformation process of seed plants from vegetative to reproductive growth (*Karlgren et al., 2011*).

Many *PEBP* family members have been identified in several plant species, including *Arabidopsis* (*Peng, Hu & Yang, 2015*), maize (*Danilevskaya et al., 2008*), cotton

Corresponding authors
Jianguang Zhang,
yyzjg@hebau.edu.cn
Yuxing Zhang, zhyx@hebau.edu.cn

(*Wang et al., 2018*), soybean (*Zhang et al., 2015*), tulip (*Leeggangers et al., 2018*), poplar (*Igasaki et al., 2008*) and kiwifruit (*Voogd et al., 2017*). In *Arabidopsis*, the *PEBP* family contains six gene members, *FT*, *TFL1*, *TWIN SISTER OF FT* (*TSF*), *BROTHER OF FT AND TFL1* (*BFT*), *ARABIDOPSIS THALIANA CENTRORADIALIS* (*ATC*) and *MFT* (*Peng, Hu & Yang, 2015*). *FT* encodes a signaling protein (or mRNA) called florigen, which serves as a long-distance signal to induce flowering (*Putterill & Varkonyi-Gasic, 2016*). Another protein encoded by *TFL1* whose sequence is closely related to *FT*, performs the opposite function to repress flowering (*Baumann et al., 2015*).

Despite extensive sequence conservation, *PEBP* genes can act as regulators of various signaling pathways to control growth and differentiation (*Chautard et al., 2004*; *Yeung et al., 1999*).

At present, identification of *PEBP* genes becomes more important in plant genomes. Pear (*Pyrus spp.*) is one of the most important fruit crops in the world. *PEBP* genes of pears have not been systematically analyzed, so that the biological functions of *PEBP* genes in pears have remained unclear until now. In this study, the *PEBP* genes in pears genomes were identified and analyzed comparatively, including gene phylogeny, chromosomal locations, protein conserved domains and *cis*-elements of their promoters. A phylogenetic analysis was also performed using *PEBP* genes searched from European pear, a wild pear and *Arabidopsis*. Moreover, we tentatively revealed the expression patterns of *PEBP* genes and functions of *PbFT* in pears during the flower bud differentiation. The results may provide a solid foundation to understand the distribution, structure and evolution of the *PEBP* genes in pears and will contribute to investigating the detailed functional differentiation and applying these genes in the future.

## MATERIALS AND METHODS

### Search for *PEBP* genes in pears

The genome sequences of "Dangshan Suli" (*Pyrus bretschneideri*), "Bartlett" (*Pyrus communis*) and "Shanxi Duli" (*Pyrus betuleafolia*) were downloaded from the Genome Database for Rosaceae (GDR) (http://www.rosaceae.org/). The *PEBP* genes information of *Arabidopsis* was downloaded from the Arabidopsis Information Resource (TAIR) (http://www.arabidopsis.org/). To identify *PEBP* genes, the PEBP-domain profile (PF01161) was obtained from the Pfam database (http://pfam.xfam.org/). The hidden Markov models (HMM) profiles were used to search against the pear protein sequence data using default parameters and removed the non-representative transcripts (*Johnson, Eddy & Portugaly, 2010*). Then, the remaining sequences were checked for the conserved PEBP domain using Pfam (*Finn et al., 2016*) and SMART (http://smart.embl-heidelberg.de/).

### Protein properties and sequence analysis

The pear PEBP sequences were uploaded to ExPASy (http://web.expasy.org/protparam/) to calculate the number of amino acids, molecular weights and isoelectric points (pI).

## Phylogenetic tree construction, gene structure and protein motif analysis

The phylogenetic tree was constructed for the sequences of PEBP, using MEGA v6.0 (*Tamura et al., 2013*) by the Neighbor-Joining (NJ) method with 1,000 bootstrap replicates. The *PEBP* gene structures were obtained by alignment of open reading frames (ORFs) with corresponding genomic sequences along with the Gene Structure Display Server (*Hu et al., 2014*). Sequence logos of domain alignments were created using the MEME (http://meme-suite.org/tools/meme) with the default settings, except that the minimum and maximum motif widths were set to 10 and 60 amino acids, respectively. PEBP of both pears and *Arabidopsis* were analyzed.

## Physical localization of *PEBP* genes on chromosomes and synteny analysis

To identify chromosomal location of *PEBP* genes, the genome annotation data was collected and mapped with TBtools (*Chen et al., 2018*). Synteny analysis of *PEBP* genes was performed by using TBtools.

## Calculation of Ks and Ka

The synonymous substitution (Ks) and non-synonymous substitution (Ka) rates were calculated using DnaSP v5 (*Librado & Rozas, 2009*), which in turn used the Nei–Gojobori (NG) method (*Li, Wu & Luo, 1985*; *Nei & Gojobori, 1986*).

## Investigation on *Cis*-elements in the promoter region

The 2,500 bp genomic sequences upstream at the transcription start site (ATG) were extracted from *Pyrus bretschneideri* genomic database, which was used to analyze putative *cis*-elements in the promoter regions of the *PEBP* genes with PlantCARE (http://bioinformatics.psb.ugent.be/webtools/plantcare/html/).

## Plant materials, growth conditions and treatment

The 7-year-old pear trees (*Pyrus bretschneideri*, cv. Xueqing) were used as the trail samples that were grown in a natural environment in Hebei Agricultural University (N 38°, E 115°).

To clarify the expression of *PEBP* genes in pears during the flower bud differentiation, leaf and apical bud samples at different developmental stages were collected at 30, 45, 60, 75, 90 and 105 days after full blooming (DAB). Meanwhile, the day-length data was obtained from the internet (https://richurimo.51240.com/). The treatment of reduction day-length on the pear leaves started from 30DAB to 120DAB in the next year. A branch of each pear tree was covered with an opaque bag from 6:30 PM until night (Fig. S1). The leaf samples were collected at 45, 75 and 105DAB in this experiment.

In order to verify the circadian rhythm of the genes, the leaf samples in the natural environment were collected from 9:00 AM to the next 6:00 AM, once for every three hours at 45, 75 and 105DAB.

*Arabidopsis* (Columbia-0) was used in this study. To generate *PbFT* overexpression transgenic plants, full-length cDNAs without the stop codon for *PbFT* were cloned into a PBI-121 vector in the sense orientation behind a cauliflower mosaic virus 35S promoter. Phenotypes of the transgenic lines were examined with T4 generation. *Arabidopsis* plants were cultured under long days (LD, light/dark, 16 h/8 h) in an artificially controlled growth room at 23 °C.

All samples that were needed to further analyze were harvested and quickly frozen in liquid nitrogen and then stored at −80 °C for further experiments. At least three independent replications were carried out for all assaying samples.

## Meristem microscopy

Apical buds of current shoots harvest at 60DAB, 75DAB and 90DAB were fixed quickly in FAA solution (70% ethanol: formalin: acetic acid = 90: 5: 5, v/v) and dehydrated in a graded ethanol series (1 h in each of 70%, 83%, 95%, 100% ethanol and 100% ethanol). Then, samples were soaked in mixtures of ethanol and xylene (1:1 v/v) 1 h and twice in absolute xylene for 1 h. Subsequently, Samples were infiltrated at 37 °C with mixtures of xylene and steedman's wax made up in the proportions of 1:1 (v/v) overnight and at 45 °C 12 h, followed by two changes of pure wax (2 h each). Specimens were transferred to kraft paper molds filled in fresh wax at 50 °C, then cooling to room temperature. Samples were sectioned into 8 μm slice and dewaxed before sealing. The slices were observed under a microscope and photographed.

## RNA extraction, cDNA synthesis and quantitative real-time PCR

Total RNA was extracted according to the manufacturer instructions of E.Z.N.A. plant RNA extraction kit (Omega Bio-tek, Norcross, GA, USA) including the gDNase. The first strand cDNA was prepared using FastQuant RT Kit (Tiangen, Beijing, China) according to the manufacturer's instructions together with gDNase.

Leaf samples harvested at 30DAB, 45DAB, 60DAB, 75DAB and 90DAB were using for transcriptome sequencing. Three independent replications were carried out for all assaying samples (since samples harvested at 30DAB and 90DAB have been confirmed as not critical samples in our research (unsubmitted), both were not set biological replicates for transcriptome sequencing). The subsequent library construction and RNA-sequencing (RNA-seq) were completed by the Shanghai Majorbio Bio-pharm Technology Co., Ltd. (Shanghai, China). The data was analyzed on the free online platform of Majorbio Cloud Platform (www.majorbio.com). The RNA-Seq data from this article can be found in the Sequence Read Archive (SRA) under accession number SRR10997902–SRR10997912.

The quantitative real-time PCR (qRT–PCR) reactions were executed using the TransStart® Top Green qPCR SuperMix (TransGen Biotech, Beijing, China). The qRT–PCR was performed using Eppendorf Mastercycler® ep realplex (Eppendorf, Hamburg, Germany). The thermal cycling conditions were 30 s at 94 °C; 44 cycles for 5 s at 94 °C, 15 s at 56 °C and 10 s at 72 °C and then performed the melting curve and cooling program. The data was calculated using the $2^{-\Delta\Delta Ct}$ (*Shi, Zhang & Chen, 2019*) method and the *PbActin* gene of *P. bretschneideri* was used as an internal control to normalize the expression

**Table 1 Characteristics of the *PEBP* genes identified in pears.**

| | Accession number | Chromosome location | Gene length (bp) | Exon number | Protein length (aa) | Molecular weight (Da) | pI |
|---|---|---|---|---|---|---|---|
| *Pyrus bretschneideri* | gene12374 | 2 | 4,134 | 4 | 174 | 19,594.15 | 7.72 |
| | gene14557 | 3 | 1,341 | 4 | 174 | 19,563.47 | 9.15 |
| | gene20820 | 5 | 1,341 | 4 | 174 | 19,593.50 | 9.15 |
| | gene4010 | 6 | 1,748 | 4 | 192 | 21,300.85 | 7.78 |
| | gene7939 | 6 | 1,781 | 4 | 192 | 21,386.94 | 7.79 |
| | gene20297 | 7 | 1,173 | 4 | 173 | 19,596.55 | 9.09 |
| | gene31860 | 11 | 1,275 | 4 | 174 | 19,643.53 | 8.86 |
| | gene23124 | 12 | 1,986 | 4 | 172 | 19,339.06 | 9.20 |
| | rna20841 | 14 | 2,677 | 4 | 172 | 19,223.00 | 8.54 |
| | gene11252 | 14 | 1,684 | 4 | 172 | 18,872.85 | 8.79 |
| *Pyrus communis* | pycom01g20860 | 1 | 890 | 4 | 173 | 19,489.22 | 8.86 |
| | pycom03g09760 | 3 | 1,330 | 4 | 174 | 19,541.42 | 8.83 |
| | pycom04g21920 | 4 | 3,602 | 4 | 174 | 19,566.14 | 7.71 |
| | pycom07g24410 | 7 | 880 | 4 | 173 | 19,566.46 | 9.09 |
| | pycom12g24180 | 12 | 1,658 | 4 | 174 | 19,520.10 | 6.72 |
| *Pyrus betuleafolia* | GWHPAAYT001914 | 1 | 865 | 4 | 173 | 19,489.22 | 8.86 |
| | GWHPAAYT040350 | 4 | 4,248 | 4 | 174 | 19,594.15 | 7.72 |
| | GWHPAAYT047043 | 6 | 1,614 | 4 | 187 | 20,607.89 | 6.50 |
| | GWHPAAYT050130 | 7 | 885 | 4 | 173 | 19,596.55 | 9.09 |
| | GWHPAAYT007785 | 11 | 1,275 | 4 | 174 | 19,643.53 | 8.86 |
| | GWHPAAYT010058 | 12 | 2,130 | 4 | 172 | 19,339.06 | 9.20 |
| | GWHPAAYT012924 | 12 | 1,934 | 4 | 174 | 19,589.21 | 7.72 |
| | GWHPAAYT019304 | 14 | 1,648 | 4 | 172 | 18,902.88 | 8.79 |
| | GWHPAAYT016784 | 14 | 2,402 | 4 | 172 | 19,223.00 | 8.54 |

levels of the target genes. Specific primers of the qRT–PCR were presented in Table S1. Three biological replications were used for the experiment.

## Statistical analysis

Statistical Product and Service Solutions v. 17.0 (SPSS, Chicago, IL, USA) was used to analyze the experimental data. *T*-test and ANOVA Tukey's multiple comparison tests were used to exam significant differences ($p < 0.05$).

## RESULTS

### Genome-wide identification of *PEBP* genes

Based on a genome-wide analysis, 24 *PEBP* genes were identified in pears, of which 10 *PEBP* genes were from *Pyrus bretschneideri* genome, 5 *PEBP* genes were from *P. communis* genome and 9 *PEBP* genes were from the *Pyrus betuleafolia* genome. The detailed information of *PEBP* genes is given in Table 1. The PEBP proteins varied more or less with
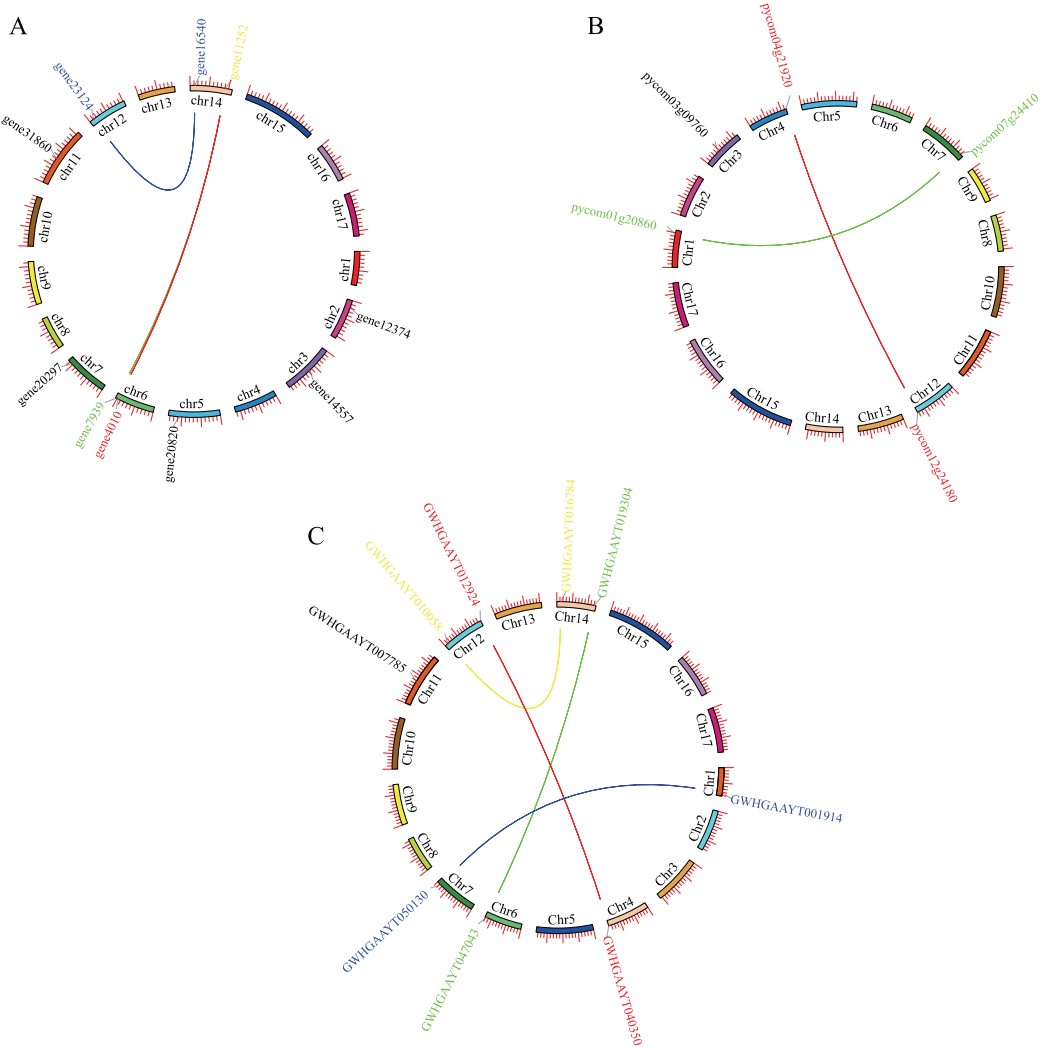

**Figure 1 Localization and synteny of the *PEBP* genes in pear genomes.** (A) *PEBP* genes in *Pyrus bretschneideri* genome. (B) *PEBP* genes in *Pyrus communis* genome. (C) *PEBP* genes in *Pyrus betuleafolia* genome. Chromosome number was indicated on the inner side of the inner circle. Gene pair with a syntenic relationship was joined by the line.

respect to characters, ranging from 172 amino acids (aa) to 192 aa in length, from 18872.85 Da to 21386.94 Da in molecular weight, and a pI from 6.5 to 9.2.

Mapping *PEBP* genes to the *P. bretschneideri* genome indicated that 10 *PEBP* genes were unevenly distributed on eight of the seventeen chromosomes, with six genes on Chr2, Chr3, Chr5, Chr7, Chr11 and Chr12, as well as two on Chr6 or Chr14. Mapping *PEBP* genes to *Pyrus communis* genome indicated that 5 *PEBP* genes were unevenly distributed on five of the seventeen chromosomes, that is, Chr1, Chr3, Chr4, Chr7, and Chr12. Mapping *PEBP* genes to *Pyrus betuleafolia* genome indicated that 9 *PEBP* genes were unevenly distributed on seven of the seventeen chromosomes, with five genes on Chr1, Chr4, Chr6, Chr7, and Chr11, as well as two on Chr12 or Chr14 (Fig. 1).
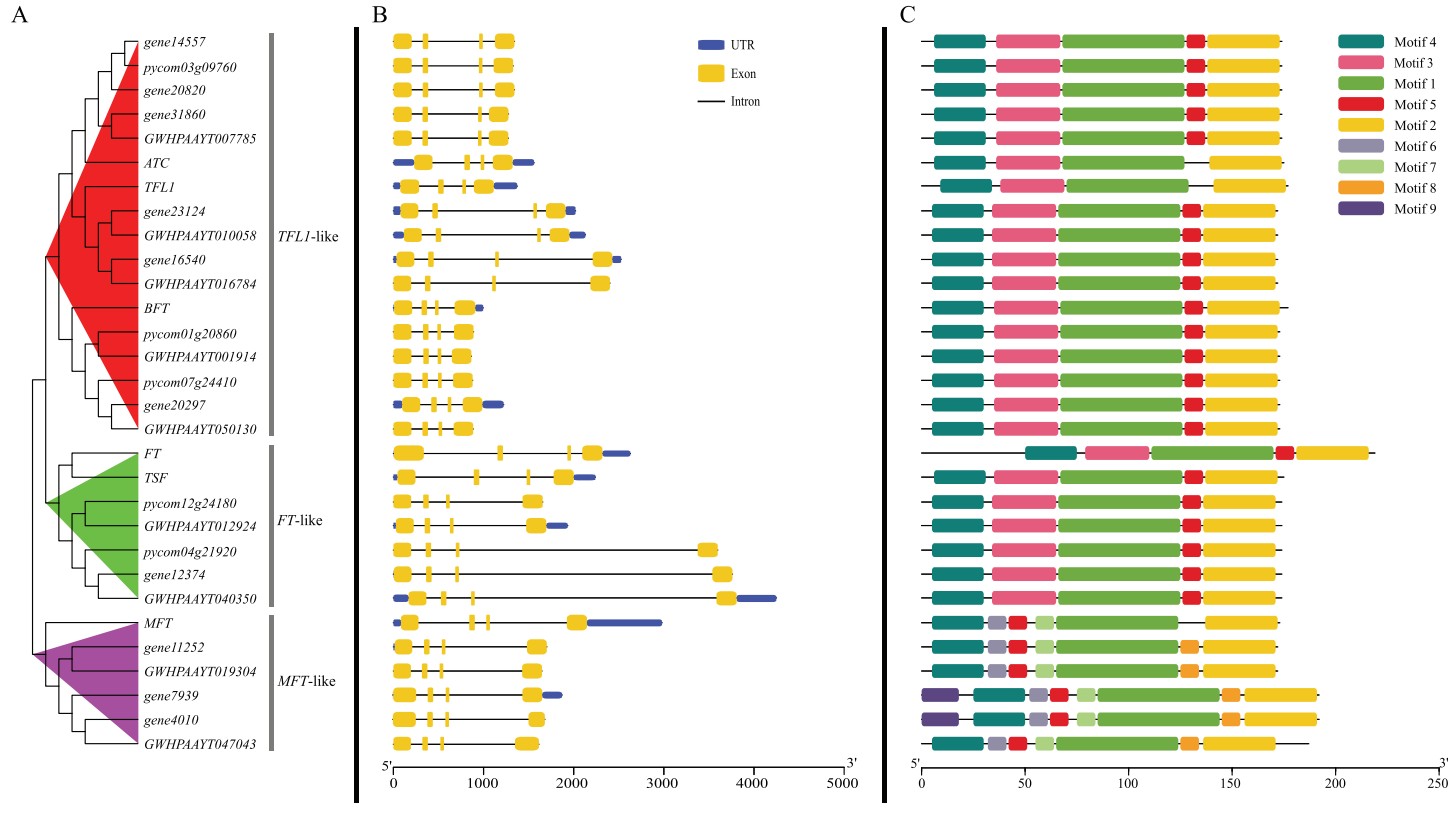

**Figure 2 Phylogenetic relationships, gene structure and motifs in *PEBP* genes from pears.** (A) The phylogenetic tree was constructed based on the full-length sequences of pear PEBP proteins using MEGA 6.0 software. (B) Exon–intron structure of pear *PEBP* genes. Blue boxes indicated untranslated 5′- and 3′-regions; yellow boxes indicated exons; black lines indicated introns. (C) The motif composition of pear PEBP proteins. The motifs, numbers 1–9, were displayed in different colored boxes. The sequence information for each motif was provided in Figs. S2–S10.

## Phylogenetic analysis of the *PEBP* family

According to the phylogenetic tree, 24 *PEBP* genes of pears were clustered into three groups (*TFL1*-like, *FT*-like and *MFT*-like), and each group contained one *Arabidopsis PEBP* gene at least (Fig. 2A). The genes, including *gene14557, gene20820, gene31806, gene23124, gene16540, gene20297, pycom03g09760, pycom01g20860, pycom07g24410, GWHPAAYT007785, GWHPAAYT010058, GWHPAAYT016784, GWHPAAYT001914* and *GWHPAAYT050130*, were clustered to *TFL1*-like group together with *ATC, BFT* and *TFL-1*. Among them, *gene23124, gene16540, GWHPAAYT010058* and *GWHPAAYT016784* were *TFL1* homologues. Additionally, these genes, including *gene12374, pycom12g24180, pycom04g21920, GWHPAAYT012924* and *GWHPAAYT040350*, were clustered to *FT*-like group together with *TSF* and *FT*. And those genes, including *gene11252, gene7939, gene4010, GWHPAAYT019304* and *GWHPAAYT047043*, were clustered to *MFT*-like group together with *AtMFT*.

## Gene structure and conservative motifs analysis

Gene structure analysis showed that all *PEBP* genes from pears contained four exons and three introns. The sizes of the second exon and the third exon were exactly 62 bp and 41 bp

**Table 2 Ka/Ks valus of *BEBP* genes between *P. betuleafolia* and *P. communis* with *P. betuleafolia*.**

| Seq_1 | Seq_2 | Ka | Ks | Ka/Ks |
|---|---|---|---|---|
| gene20297 | pycom01g20860 | 0.058323 | 0.255761 | 0.228038 |
| gene20297 | pycom07g24410 | 0.002747 | 0.017107 | 0.160556 |
| gene4010 | GWHPAAYT019304 | 0.018274 | 0.127154 | 0.143714 |
| gene7939 | GWHPAAYT019304 | 0.069975 | 0.177029 | 0.395272 |
| gene4010 | GWHPAAYT047043 | 0.028379 | 0.033548 | 0.845944 |
| gene20297 | GWHPAAYT001914 | 0.058323 | 0.255761 | 0.228038 |
| gene20297 | GWHPAAYT050130 | 0 | 0.008547 | 0 |
| gene23124 | GWHPAAYT010058 | 0 | 0.007937 | 0 |
| gene23124 | GWHPAAYT016784 | 0.053811 | 0.100288 | 0.536559 |
| gene16540 | GWHPAAYT010058 | 0.053332 | 0.091653 | 0.581889 |
| gene11252 | GWHPAAYT019304 | 0.002585 | 0.023717 | 0.108996 |
| gene16540 | GWHPAAYT016784 | 0 | 0.007813 | 0 |
| gene11252 | GWHPAAYT047043 | 0.044601 | 0.178034 | 0.250522 |

(Fig. 2B). Nine motifs (Figs. S2–S10) in pear PEBPs were detected and all pear PEBP proteins contained motifs 1–5. Motif 5 contained D-P-D-x-P and G-x-H-R residues, which were conserved residues to the *PEBP* family of all plants. However, only *MFT*-like group proteins contained motifs 6–9 (Fig. 2C).

## Selective pressure analysis

The gene pairs were obtained by synteny analysis. The ratio of non-synonymous substitutions (Ka)/synonymous substitutions (Ks) was evaluated to detect the modes of selection of *PEBP* genes. Generally, Ka/Ks > 1 indicated a positive selection with an accelerated evolution; Ka/Ks < 1 indicated functional constraints with a negative selection; and Ka/Ks = 1 signified a neutral selection. The results showed that the Ka/Ks ratios of 13 duplicated *PEBP* gene pairs ranged from 0.11 to 0.85 with an average of 0.27 (Table 2). Additionally, the majority of the ratios were less than 0.39, suggesting that the duplicated *PEBP* genes of pears had mainly undergone strong purifying selection, with the slowly evolving at the protein level of this gene family.

## *Cis*-elements in the *PEBP* genes promoters of *Pyrus bretschneideri*

We analyzed 2,500 bp genomic sequences upstream at the transcription start site (ATG) of *Pyrus bretschneideri PEBP* genes. About 50 putative *cis*-elements were identified in each promoter of *PEBP* genes that were classified into three groups, including hormone responses, plant growth and development and biotic/abiotic stress responses (Fig. 3). There were numerous light-responsive elements on these promoters, such as AE-box, Box 4, G-box, I-box, and GT1-motif. The promoter of *gene12374* possessed three Auxin-responsive elements and one GA-responsive element, and the promoter of *gene23124* had one Auxin-responsive element, which was not found in the promoters of other *Pyrus bretschneideri PEBP* genes.

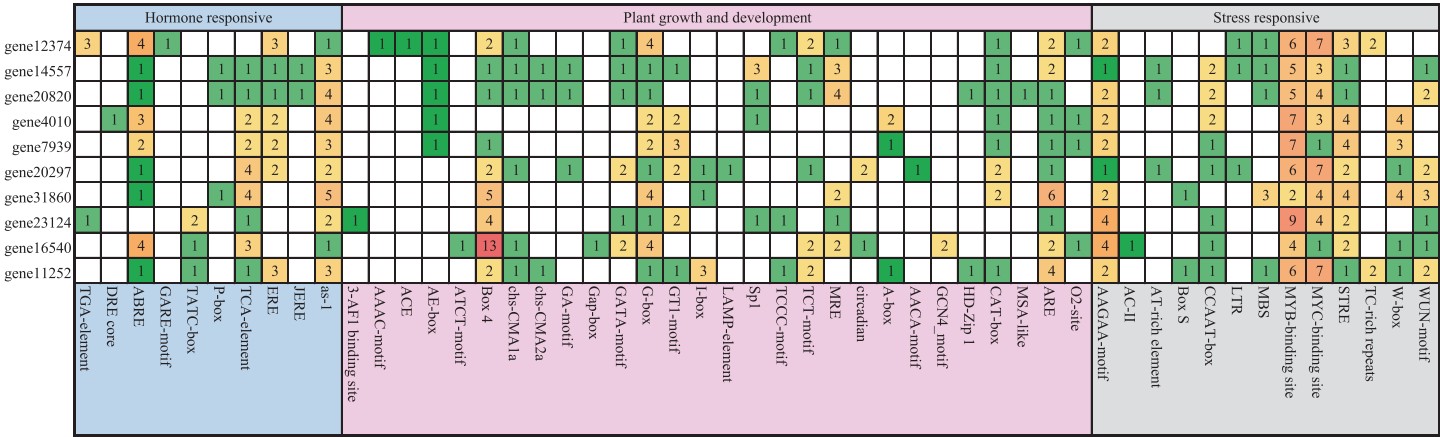

**Figure 3** Investigation of *cis*-acting element numbers in the *PEBP* Genes Promoters of *Pyrus bretschneideri*. The different colors and numbers of the grid indicated the numbers of different promoter elements in these *PEBP* genes.

## Expression of *PEBP* genes during the flower bud differentiation

Expression of pear *PEBP* genes obtained from the RNA-seq data of "Xueqing" leaves at different stages was analyzed (Table S2). It was found that just one gene (*gene12374*) was expressed in all five development stages and was relatively highly expressed in leaves (Fig. 4A). The other genes had low or no expression. Phylogenetic analysis revealed that the closest relationship between *gene12374* and *Arabidopsis PEBP* gene was *FT*, so *gene12374* was re-named *PbFT*. *PbFT* expression patterns with the qRT–PCR experiment were shown in Fig. 4B. The expression trends of *PbFT* were similar to RNA-seq data. The highest expression of *PbFT* was consistently observed at 75DAB. Then, the day-length of a period from 30DAB to 120DAB was recorded (Fig. 4C). According to the microscopic observation of current shoot apical meristems, it was not found abnormal at 60DAB. This period was maybe in the flower bud physiological differentiation stage. Whereas, it was found that the flower bud morphological differentiation stage had already entered at 75DAB (Figs. 4D–4F). Meanwhile, the expressions of *PbFT* was also the highest in apical buds (Fig. 5A). Those findings suggested that the period from 60DAB to 75DAB was critical for the flower bud differentiation, and the expression of *PbFT* was increasing strongly during this period. Additionally, the expressions of two *TFL1* homologues (*gene23124* and *gene16540*) were higher before 60DAB in apical buds. The lowest expressions of *gene23124* and *gene16540* occurred at 75DAB. Subsequently, the expression of *gene23124* remained at a lower level, but the expression of *gene16540* increased gradually (Figs. 5B and 5C).

Interestingly, the time of the highest expression of *PbFT* was found around the day with the longest daylength. Subsequently, the treatment of day-length reduction on the pear leaves showed that if the day-length was reduced, the expression of *PbFT* also decreased (Fig. 6). The expression of *PbFT* under the natural condition was up to twice as much as under the treatment of day-length reduction at 75DAB.

On the other hand, the expression of *PbFT* was at a high level during the day, but it dropped suddenly as night fell and remained at a lower level until dawn (Fig. 7).

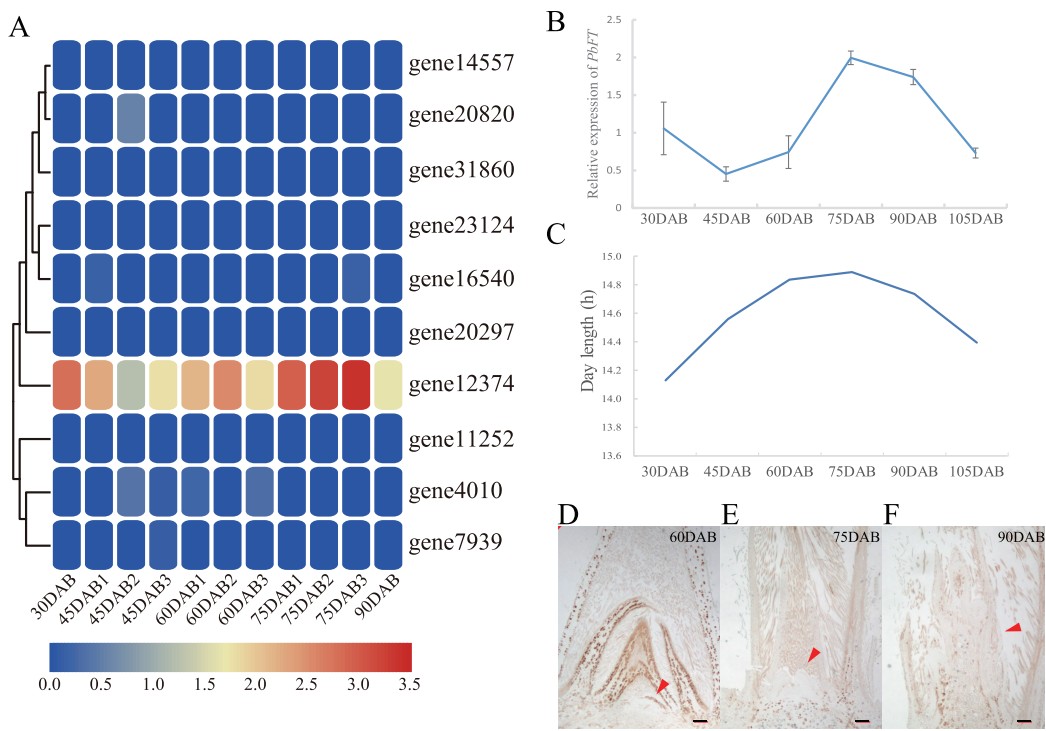

**Figure 4 Expression profiles of *PEBP* genes in "Xueqing" leaves during different stages.** (A) Heatmap of *PEBP* genes expression. (B) Relative expression of *PbFT* in "Xueqing" leaves during different stages. (C) The day-length of a period from 30 DAB to 105DAB. Microscopy of current shoot apical meristems at (D) 60DAB, (E) 75DAB and (F) 90DAB. Red arrows pointed to apical meristems. Scale bar, 100 μm.

## Overexpression of *PbFT* in relation to acceleration of flowering time

To investigate the role of *PbFT* in flower bud induction regulation, *PbFT* was overexpressed under the control of the 35S promoter in the wild-type *Arabidopsis* (Col-0). A total of nine independent lines of overexpressing *PbFT* (PbFT-OE) plants that showed obviously enhanced expression of *PbFT* were obtained. Two independent lines were randomly selected for further analysis. The transgenic lines were confirmed by PCR with the 35S forward primer (35S-F) and *PbFT*-specific reverse sequencing primer (PbFT-R) (Table S1). The days from germination to flowering was 32.11 for WT plants, but 24.33 and 25.33 with PbFT-OE3 and PbFT-OE9 lines. The corresponding total rosette leaf number was 18.29, 8.57 and 10.17 in WT, PbFT-OE3 and PbFT-OE9 lines. The days from germination to flowering were significantly accelerated and the total rosette leaf number decreased significantly in the PbFT-OE lines compared with the WT plants, but two overexpression lines showed no significant difference in the relative parameters (Fig. 8). The diameters of the transgenic lines PbFT-OE3 and PbFT-OE9 main stem were 0.76 and 0.82 mm, which were significantly larger than 0.5 mm of WT. In the absence of human interference, the transgenic plants were lodging, as opposing to the wild-type upright phenotype (Fig. 8).

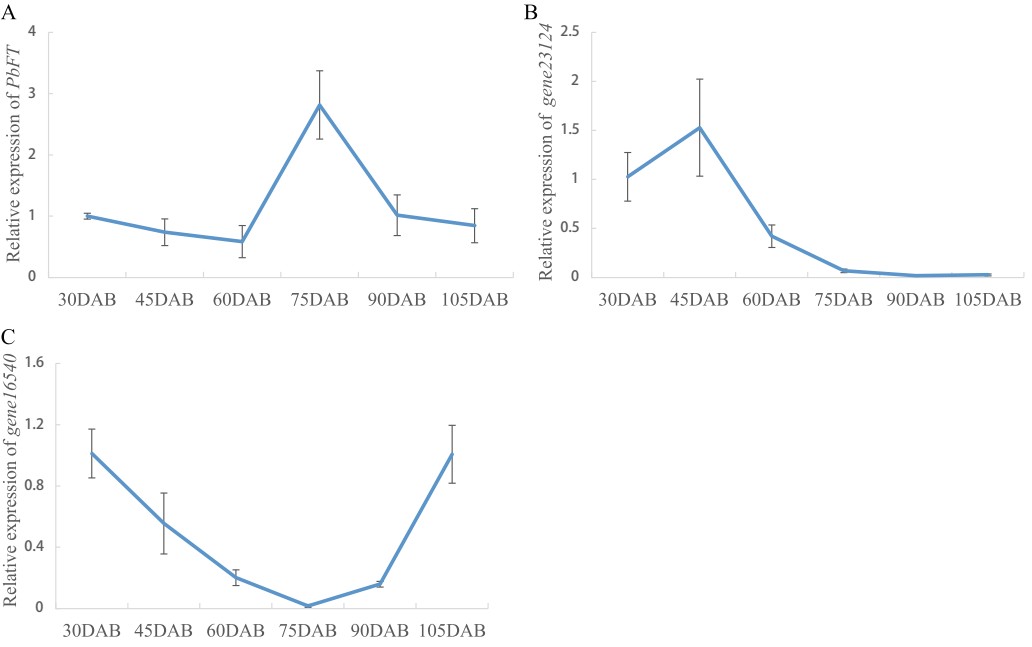

**Figure 5 Relative expressions of *PbFT* and *TFL1* homologues in "Xueqing" buds during different stage.** Relative expression of (A) *PbFT*, (B) *gene23124* and (C) *gene16540*.

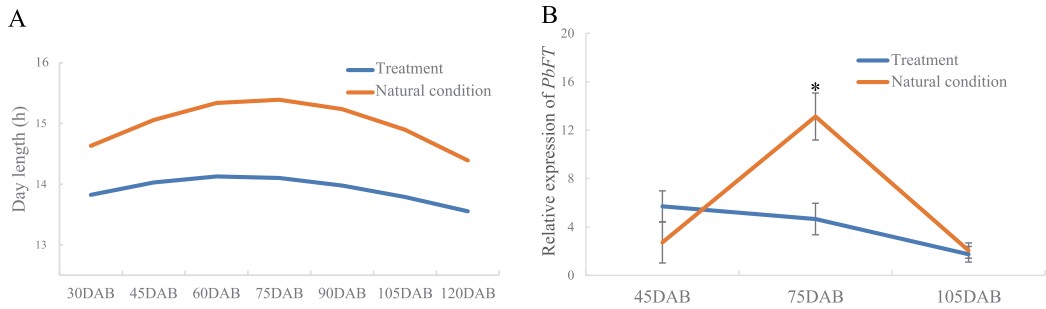

**Figure 6 The relationship between relative expression of *PbFT* and day-length.** (A) Comparison the day-length between the treatment and natural condition. (B) Comparison the relative expression of *PbFT* between under the treatment and natural condition. Results were presented as mean ± SE. ($n = 3$, *$p < 0.05$).

## DISCUSSION

### Conservation and evolution of PEBP genes in pears

So far, the genome sequencing of "DangshanSuli" (*Pyrus bretschneideri*), "Bartlett" (*Pyrus communis*) and "Shanxi Duli" (*Pyrus betuleafolia*) has been completed. These existing information resources can be used to analyze the evolution and gene function of the *PEBP* family from the perspective of bioinformatics. The PEBPs were a kind of evolutionarily conserved proteins. As the *PEBP* genes of other species like *Arabidopsis* and apple (*Malus domestica*), the *PEBP* genes of pears consisted of four exons and three introns and meanwhile the second exon contained 62 bases and the third exon contained 41 bases.

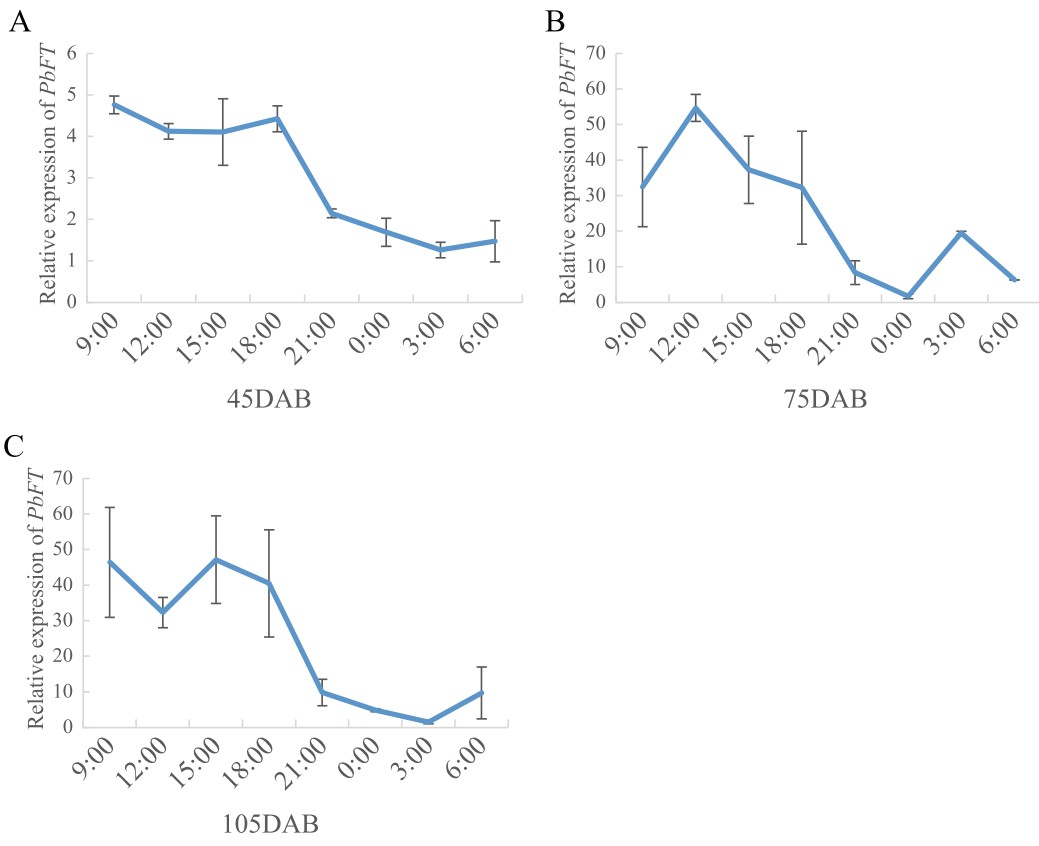

**Figure 7 The circadian pattern of *PbFT* expression.** Relative expression of *PbFT* in (A) 45DAB, (B) 75DAB and (C) 115DAB.

In addition, the Ka/Ks of *PEBP* gene pairs obtained from collinearity analysis in different genera were far less than 1. This result indicated that the *PEBP* genes of pears have undergone purifying selections, which also confirmed that the *PEBP* family was quite conservative. However, the number of *PEBP* genes and their distribution on chromosomes of *Pyrus bretschneideri*, *Pyrus communis* and *Pyrus betuleafolia* were different in this study. Ten and nine genes from *Pyrus bretschneideri* and *Pyrus betuleafolia* were identified, while only five genes were identified from *Pyrus betuleafolia*. This result suggested that different degrees of gene loss had occurred during the evolution of *Pyrus*, especially in the *Pyrus communis* genome, at least in the *PEBP* family of "Bartlett" pear. Interestingly, among the materials used for genome sequencing, "DangshanSuli" (*Pyrus bretschneideri*) and "Shanxi Duli" (*Pyrus betuleafolia*) belonged to Asian pears and "Bartlett" (*Pyrus communis*) was a European pear. This situation might be due to the independent domestication processes for either Asian or European pears (*Wu et al., 2018*).

In addition, more attention should be paid to the genes in the *FT*-like group. *FT* and *TWIN SISTER OF FT* (*TSF*) had redundant functions in promoting flowering (*Jin et al., 2015*; *Yamaguchi et al., 2005*), and they had five ortholog genes in pears. There were two genes belonging to both "Bartlett" and "Shanxi Duli", but just one, *PbFT*, was

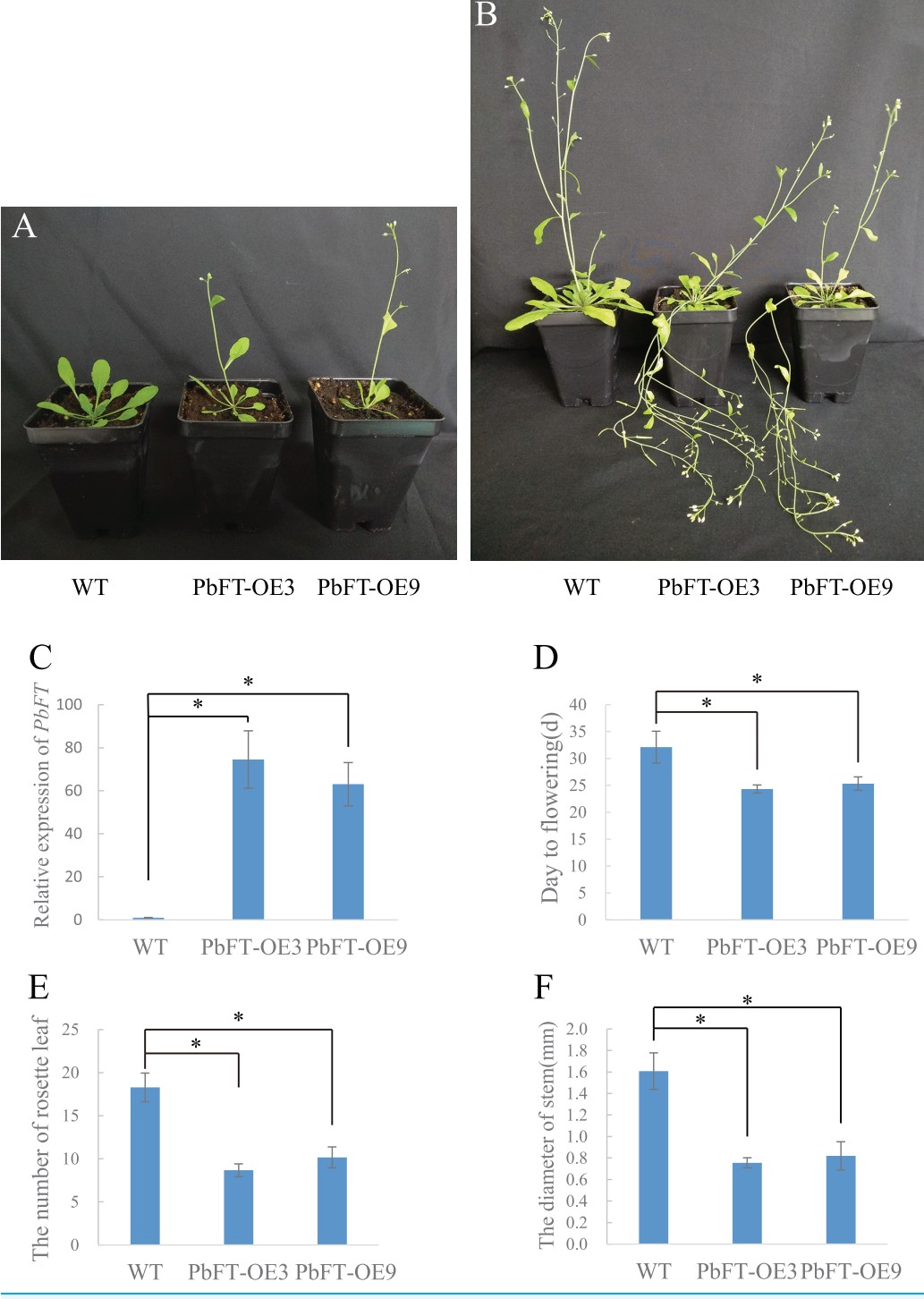

**Figure 8 Flowering phenotype of *35S:PbFT* transgenic *Arabidopsis*.** Representative images of transgenic *Arabidopsis* showing their flowering phenotype at (A) 30d and (B) 60d after germination. Flowering phenotype of overexpressing *PbFT* transgenic *Arabidopsis* assessed by (C) the expression of *PbFT*, (D) days from germination to flowering, and (E) rosette leaf number. (F) The diameters comparison of overexpressing PbFT transgenic *Arabidopsis* and WT. Results were presented as mean ± SE ($n \geq 3$, *$p < 0.05$, Students *t*-test).

identified in "DangshanSuli" genome. Since *PbFT* was so closely related to the *AtFT* and *AtTSF*, we speculated that it may also be similar in functions.

## Expression and function of *PbFT*

*FT* and *TFL1* played important roles in flowering induction (*Gao et al., 2017*; *Putterill & Varkonyi-Gasic, 2016*). The expressions of *PbFT* and *TFL1* homologues (*gene23124* and *gene16540*) in pear buds showed the opposite expression patterns before and in the critical period of the flower bud differentiation. However, it was found that only *PbFT* among pear *PEBP* genes was relatively highly expressed in leaves. It suggested that *PbFT* played positive roles both in leaves and buds, but *gene23124* and *gene16540* played negative roles only in buds. In *Arabidopsis*, *TFL1* expression was merely limited to shoots (*Baumann et al., 2015*). This was similar to our results. It suggested that *gene23124* and *gene16540* had the opposite function to *PbFT*. As a receptor of the plant, the leaves can directly sense the changes of the external environment, especially the time of day-length, so as to regulate the growth and development of the plant through corresponding signal transduction. FT protein is synthesized in the leaf vasculature and transported to the apical meristem of a shoot through the phloem (*Corbesier et al., 2007*; *Lin et al., 2007*; *Tamaki et al., 2007*; *Zeevaart, 2008*). By analyzing the promoter, it was found that there were typical light-responsive elements, AE-box, Box 4, G-box and TCT-motif, on the *PbFT* promoter. This suggested that the expression of *PbFT* may be regulated by light. According to our transcriptome data, only the *PbFT* gene, among the *PEBP* genes of *Pyrus bretschneideri*, had a relatively high expression level in the leaves. It was found that the expression of *PbFT* was increasing strongly during critical period, which was from the flower bud physiological differentiation stage to morphological differentiation stage and the highest expression of *PbFT* was observed at 75DAB, when was about the longest daytime of a year. Shortening the day-length of some branches indicated that the expression of *PbFT* was down-regulated. This was a good indication that the expression of the *PbFT* gene was related to the flower bud induction and considerably regulated by the day-length in a year.

Many scholars have concluded that *FT* is strongly expressed around dusk in LD in the incubator (*Mouradov, Cremer & Coupland, 2002*; *Song, Ito & Imaizumi, 2013*). However, we found that *PbFT* gene remained at a high expression level during the day and at a lower level in the evening until dawn. *PbFT* expression showed an unusual circadian rhythm. As the temperature and light in the incubator are usually constant but greater changes exist in the natural environment, these may result in a different pathway of regulation mechanism with *PbFT*.

At least six signal pathways (photoperiod, autonomous, age, gibberellin, vernalization, ambient temperature) that regulate the flowering process have been demonstrated (*Blumel, Dally & Jung, 2015*; *Kim et al., 2009*; *Mutasa-Gottgens & Hedden, 2009*; *Srikanth & Schmid, 2011*; *Wang, 2014*). As a mobile signal florigen gene, in several pathways, FT can initiate the floral induction in some species (*Blumel, Dally & Jung, 2015*; *Cao et al., 2015*; *Corbesier et al., 2007*; *Liu et al., 2013*; *Oda et al., 2012*; *Pin & Nilsson, 2012*; *Tamaki et al., 2007*). The research on *FT* genes has been progressed in many horticultural plants (*Wilkie, Sedgley & Olesen, 2008*). However, the role of the *PbFT* during flower bud

differentiation remains unclear in pears. In this study, it was found the expression of *PbFT* increased during the period of floral initiation. Moreover, the overexpression of *PbFT* dramatically accelerated the starting time for flowering *Arabidopsis*. This implies that the *PbFT* might play a role in promoting flower bud differentiation in pears.

Additionally, homologs of *FT* genes in crops have pleiotropic functions that can mediate numerous developmental processes, such as growth, plant architecture control, fruit set and tuber formation (*Pin & Nilsson, 2012*). *AcFT1* promotes bulb formation, while *AcFT4* inhibits bulb production in onions (*Lee et al., 2013*). Tuberization is regulated by the *FT* homolog *StSP6A* through modulation of source-sink in potatoes (*Abelenda et al., 2019*; *Lehretz et al., 2019*). Our transgenic plants were thinner than WT, which was caused by overexpression of *PbFT*. This finding suggests that *PbFT* has a role in not only promoting flower bud differentiation, but also regulating the balance between vegetative and reproductive growth.

## CONCLUSIONS

We identified 10, 5 and 9 *PEBP* genes from genomes of *Pyrus bretschneideri*, *Pyrus communis* and *Pyrus betuleafolia*, respectively. The PEBPs is a kind of evolutionarily conserved proteins. However, the number of *PEBP* genes and their distribution on chromosomes of *Pyrus bretschneideri*, *Pyrus communis* and *Pyrus betuleafolia* are different. This situation may be due to the independent domestication processes for either Asian or European pears. Moreover, we determined that *PbFT*, which can be regulated by day-length and circadian clock, possessed the function of promoting flower bud differentiation in pears. Additionally, *PbFT* may play a role in regulating the balance between vegetative and reproductive growth.

## ACKNOWLEDGEMENTS

We sincerely thank Professor Jiafu Jiang (Nanjing Agricultural University) for providing the *Arabidopsis* seeds.

### Funding

This research was funded by the Agriculture Research System of China (CARS-28) and the Graduate Student Innovation Fund Project in Hebei Province (CXZZBS2020088). The funders had no role in study design, data collection and analysis, decision to publish, or preparation of the manuscript.

### Grant Disclosures

The following grant information was disclosed by the authors:
Agriculture Research System of China: CARS-28.
Graduate Student Innovation Fund Project in Hebei Province: CXZZBS2020088.

### Competing Interests

The authors declare that they have no competing interests.

## Author Contributions

- Shuliang Zhao conceived and designed the experiments, performed the experiments, analyzed the data, prepared figures and/or tables, authored or reviewed drafts of the paper, and approved the final draft.
- Yarui Wei conceived and designed the experiments, performed the experiments, prepared figures and/or tables, and approved the final draft.
- Hongguang Pang conceived and designed the experiments, performed the experiments, prepared figures and/or tables, and approved the final draft.
- Jianfeng Xu conceived and designed the experiments, prepared figures and/or tables, and approved the final draft.
- Yingli Li conceived and designed the experiments, prepared figures and/or tables, and approved the final draft.
- Haixia Zhang conceived and designed the experiments, prepared figures and/or tables, and approved the final draft.
- Jianguang Zhang conceived and designed the experiments, prepared figures and/or tables, authored or reviewed drafts of the paper, and approved the final draft.
- Yuxing Zhang conceived and designed the experiments, prepared figures and/or tables, and approved the final draft.

## Data Availability

Data is available at NCBI SRA: SRR10997902, SRR10997903, SRR10997904, SRR10997905, SRR10997906, SRR10997907, SRR10997908, SRR10997909, SRR10997910, SRR10997911, SRR10997912.

## Supplemental Information

Supplemental information for this article can be found online at http://dx.doi.org/10.7717/peerj.8928#supplemental-information.

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
