# Peer review of "Genome-wide identification of the PEBP genes in pears and the putative role of PbFT in flower bud differentiation"

_PeerJ, doi:10.7717/peerj.8928_

## Round 0.1 · original submission · Major Revisions

In the present paper authors describe the identification of PEPB family members from the pear genome, phylogenetic analysis and investigate the expression of a set of 10 of these in fruit development. In addition, the authors analyzed the function of a candidate member. The initial intention of the paper is of a certain theoretical significance.

However, the following points must be addressed:
1. ‘Cis-’ requires italics.
2. The analysis of the Selective Pressure Analysis Section needs to be expanded.
3. 35s:FT, Pb-OE, PbFT-OE should be unified. E.g. Fig 7.
4. References are outdated and need to be updated.
5. SRA numbers for the transcriptome are required.

Reviewer 1 ·

Basic reporting

.

Experimental design

.

Validity of the findings

.

Additional comments

Review PeerJ #44784

The authors tried to clarify the role of PEBP gene families, especially FT gene, and they suggested that pear FT genes was regulated by day length and circadian clock. Moreover, transgenic arabidopsis over-expressing FT, suggested that FT played a role in not only promoting flower bud differentiation, but also regulating the balance between vegetative and reproductive growth. The former parts on bioinformatics were also important but not so interesting; rather latter parts on physiological analysis should be more strengthen. Accordingly, I felt, direct results were not provided in the current manuscript for supporting the authors’ conclusion. The authors should provide the status during flower bud development including initiation/induction stages, along with the expression data. Without the flower bud status, it was impossible to get a authors’ conclusion.
Other suggestions:
1) Provide the similar data to FT on TFL1.
2) In pear, the movement of FT from leaves to meristems may not occurr because the grafting of wild scions of European pear on the FT-over-expressed pear rootstocks did not cause the early flowering phenotypes of the grafted wild scion cultivars. Therefore, the leaves as PCR materials for FT expressions may not be adequate. I recommend the use of buds instead of leaves.

Reviewer 2 ·

Basic reporting

1- The writing of the manuscript is quite coarse, and the authors may use professional editing service to improve the logic flow and accuracy of the manuscript. Please rewrite the Introduction section. Specifically, rearrange the logic between paragraphs, and add more new references (recent research progress).

2- Please rewrite the Abstract section. The highlights of the study were not highlighted.

3- Journals often require both the manufacturer's name and location (city, state, country) for specialized equipment, software, and reagents. .

4- Please ensure that all gene and protein names are formatted consistently throughout the manuscript and adhere to the appropriate conventions. E.g. ‘PEBP’ in PEBP genes needs italics.

5-Abbreviations and acronyms are often defined the first time they are used within the abstract and again in the main text and then used throughout the remainder of the manuscript. E.g. FT.

Experimental design

1.The statistical analysis was not appropriate as it was not provided by the authors. How many treatments were used? How the data was analyzed?

2.How many replicates do you use for your transcriptome? The transcriptomic analysis is based on your replicates of samples among different treatments. You should highlight how many replicates do you test and show the repeatability error in Results section to make sure this experiment is reliable. The content of transcriptome sequencing needs to be described in detail in the method.

Validity of the findings

The overall concept of the research and the experimental design are correct.

Additional comments

I carefully read the submission titled ‘Genome-wide identification of the PEBP genes in
pears and the putative role of PbFT in flower bud differentiation’. The manuscript would be of general interest; however, the some points must be addressed.
I do accept this paper for further revision after these topics are corrected/clarified.

---

## Round 0.2 · Major Revisions

We have received a reports from Reviewer 1 on your manuscript "Genome-wide identification of the PEBP genes in pears and the putative role of PbFT in the flower bud differentiation".

As you can see, they still have concerns, and in fact they are recommending rejection. Below, please find the comments for your perusal an response.

Reviewer 1 ·

Basic reporting

see below

Experimental design

see below

Validity of the findings

see below

Additional comments

The authors tried to improve the manuscript, but I did not satisfy their revision as indicated the below reasons.
1) They did not provide the expression data on TFL1. It was strange that the expression of TFL1 homologue did not change so much in Fig. 4A. Usually, TFL1 expression could decrease before bud differentiation (=before the increase in FT expression). Discuss the reason(s) for no-expression on TFL1 homologue; Pear TFL1 did not express in the leaves so that I asked to carry out expression analysis of FT and TFL1 in the apical buds during bud differentiation as pointed out in my first review.
2) In line 222, the authors used “maybe”, which means that there are no direct data showing the expression regulations of FT through day-length/circadian clock. Therefore, provide direct evidences on their relationships.
3) Provide the reason for focusing the Pyrus bretchneideri from Figure 3.
4) In Fig. 4A, change to XA, XB・・・・, to 30 DAF, 45 DAF.
5) In Table 1, distinguish the pear species by the horizontal lines.
6) In line 218, “The expression patterns with the qRT-PCR experiment were shown in Figure 2C”. Fig. 2 is correct?

---

## Round 0.3 · accepted · Accept

I carefully read the submission titled ‘Genome-wide identification of the PEBP genes in pears and the putative role of PbFT in flower bud differentiation’. The manuscript would be of general interest; however, the following points from R3 must be addressed (this can be done while in production)

Reviewer 2 ·

Basic reporting

The author has done the revision better.

Experimental design

no comment

Validity of the findings

no comment

Additional comments

1. Punctuation and formatting errors persist. Need to be perfected. E.g: ‘assaying samples. (Since samples harvested at 30DAB and 90DAB have been confirmed as not critical samples in our research (unsubmitted), both’.

2. What does XA, XB, XC, XD and XE stand for?

3.Table 1-Capitalize the first letter of ‘gene length’.

The format of the article needs further specification.

Reviewer 3 ·

Basic reporting

This MS entitled 'Genome-wide identification of the PEBP genes in pears and the putative role of PbFT in the flower bud differentiation' reported the isolation and the functional research of PbFT in pear. They found that PbFT might play a role in not only promoting flower bud differentiation, but also regulating the balance between vegetative and reproductive growth.

Experimental design

The experiments were well designed.

Validity of the findings

This research provides important data for phase transition in pears.

Additional comments

This MS entitled 'Genome-wide identification of the PEBP genes in pears and the putative role of PbFT in the flower bud differentiation' reported the isolation and the functional research of PbFT in pear. They found that PbFT might play a role in not only promoting flower bud differentiation, but also regulating the balance between vegetative and reproductive growth.
It is qualified to be published in Peer J after minor revision:
1 In the abstract, 'A trail of different treatments indicated that PbFT expressions in leaves were regulated by day-length and circadian clock, which was completely different from many other scholars’ findings' could be changed to ' PbFT expressions in leaves were firstly found to be regulated by day-length and circadian clock'.
2 The format of the MS should be modified, there should be spaces in the second paragragh of each parts.
3 Discuss the possible role of PbFT in phase transition.